# Manufacture of Contact Lens of Nanoparticle-Doped Polymer Complemented with ZEMAX

**DOI:** 10.3390/nano10102028

**Published:** 2020-10-15

**Authors:** Lina M. Shaker, Ahmed A. Al-Amiery, Abdul Amir H. Kadhum, Mohd S. Takriff

**Affiliations:** 1Laser and Optoelectronics Engineering Department, University of Technology, Baghdad 10001, Iraq; 2Energy and Renewable Energies Technology Center, University of Technology, Baghdad, Baghdad 10001, Iraq; dr.ahmed1975@gmail.com; 3Department of Chemical & Process Engineering, Faculty of Engineering & Built Environment, Universiti Kebangsaan Malaysia, Bangi, Selangor 43600, Malaysia; amir8@ukm.edu.my (A.A.H.K.); sobritakriff@ukm.edu.my (M.S.T.)

**Keywords:** PMMA-TiO_2_, contact lens, vision correction, high refractive index, modulation transfer function, image simulation

## Abstract

Many people suffer from myopia or hyperopia due to the refractive errors of the cornea all over the world. The use of high refractive index (RI), Abbe number (*ν_d_*), and visible light transmittance (T%) polymeric contact lenses (CLs) holds great promise in vision error treatment as an alternative solution to the irreversible laser-assisted in situ keratomileusis (LASIK) surgery. Titanium dioxide nanoparticles (TiO_2_ NPs) have been suggested as a good candidate to rise the RI and maintain high transparency of a poly(methyl methacrylate) (PMMA)-TiO_2_ nanocomposite. This work includes a preparation of TiO_2_ NPs using the sol gel method as well as a synthesis of pure PMMA by free radical polarization and PMMA-TiO_2_ CLs using a cast molding method of 0.005 and 0.01 *w*/*v* concentrations and a study of their effect on the aberrated human eye. ZEMAX optical design software was used for eye modeling based on the Liou and Brennan eye model and then the pure and doped CLs were applied. Ocular performance was evaluated by modulation transfer function (MTF), spot diagram, and image simulation. The used criteria show that the best vision correction was obtained by the CL of higher doping content (*p* < 0.0001) and that the generated spherical and chromatic aberrations in the eye had been reduced.

## 1. Introduction

It remains a challenging task to evolve a polymer that fulfills all the required features for contact lenses (CLs) applications simultaneously. There has been continual evolution in the CL materials field since these materials were invented. Fundamentally, CLs have been classified into hard, soft, and rigid gas permeable (RGP) according to their elasticity. Even though hard CLs are longer lasting than others, these lenses tend to loss their popularity. Hard CLs are primarily based on hydrophobic materials such as poly(methyl methacrylate) (PMMA), whereas soft CLs are made of biocompatible hydrogels [1].

Recent advances in nanoscience and nanotechnology [2] have facilitated the sciences to develop new polymers hybridized with high refractive index (RI) nanoparticles (NPs). Typically, silicone-hydrogel [3], poly(vinyl alcohol) [4] (PVA) CLs [5], and other plastic polymers [6], in addition to manufacturing techniques, are used to produce transparent (T > 90%), lightweight, and impact-resistant CLs [7]. RGP CLs are expensive and suffer from a lack of hydrophilic monomers, but they are more flexible than PMMA CLs due to their integration with low modulus components and high efficiency in reducing generated aberrations [8].

In general, all soft CLs significantly and adversely affect the tear physiology by reducing tear thinning time and increasing the evaporation rate [9]. All the CL materials discussed above are classified as polymers. To suit optometric applications, the polymer material must be biocompatible, transparent, and able to combine high water content, good mechanical strength, and high refractive index (*n*) and it must have low dispersion (*ν_d_*) to allow optical correction of refractive errors. In this regard, hydrogels have good biocompatibility; however, their mechanical weakness characteristic due to their high water content limits their practical applications [10].

Nevertheless, hydrogel materials with high water content typically have a low n factor and may cause light dispersion. This is undesirable for the purpose of vision correction because lenses with low index of refraction materials require a relatively high thickness to achieve the required refractive power. An effective way to increase polymers’ refractive index is by introducing high n inorganic nanoparticles into the organic polymers. Recently, doping with TiO_2_ [11,12], ZnO [13,14], ZnS [15], ZrO_2_ [16], Al_2_O_3_ [17] NPs, and so on, has been utilized to obtain nanocomposites of high optical quality for nanomaterial applications [18,19]. These nanocomposites can be exploited in CL manufacturing.

The aim of this work was to prepare pure PMMA and PMMA-TiO_2_ CLs with different TiO_2_ NP contents. The ZEMAX optical design program is used to evaluate and model the optics of the prepared CLs in comparison with an aberrated human eye. The modulation transfer function (MTF) and image simulation have better assisted us in image analysis.

## 2. Analysis Criteria

### 2.1. Modulation Transfer Function

The MTF considers the contrast degradation that occurs in sinusoidal patterns of spatial frequency, or, rather, is the ratio of image contrast to object contrast at all spatial frequencies. Spatial frequency, which measures the capabilities of the human visual system, was examined. The contrast (modulation) of a sinusoidal pattern is defined as [20]
(1)$$MTF=\frac{I_{Max}-I_{Min}}{I_{Max}+I_{Min}}$$
where *I_max_* is the irradiance of the peak of the sinusoid and *I_min_* is the irradiance of the trough of the sinusoid. At a certain value of spatial frequency the *MTF* will be zero; this spatial frequency value is called the cutoff frequency (*v_cut off_*) (measured in cycles/mm in this work) and is given by [21]
(2)$$v_{cutoff}=\frac{1}{\lambda (F/\#)}$$
where *F*/*#*, i.e., *F*/number of an optical system, refers to the ratio of the lens focal length (*F*) to the pupil diameter (PD).

### 2.2. Root Mean Square (RMS)

A spot diagram is a way of visualizing the aberration effect which is had on image quality and hence lens resolution. *RMS* refers to the root mean square of the spot in the image plane. It is calculated as the *RMS* of all distances between each peripheral intersection (*x_i_*,*y_i_*) with the image plane and a reference point (*x*_0_,*y*_0_) generated by intersection of the chief ray. *RMS* is computed from Equation (3) [22], i.e.,
(3)$$R_{RMS}=\frac{\sum_{i=1}^{n}{\left(x_{i}-x_{0}\right)}^{2}+{\left(y_{i}-y_{0}\right)}^{2}}{n}$$

## 3. Materials and Methods

PMMA polymer was prepared using free radical polymerization (FRP) and TiO_2_ NPs were prepared using the sol gel method [23]. A solution casting method was used to prepare CLs with different concentrations of TiO_2_ NPs.

### 3.1. PMMA Preparation

Materials used for the PMMA polymer preparation were methyl methacrylate monomer (MMA) (C_5_H_8_O_2_), which was obtained from Ruby Dent, tetrahydrofuran (THF) (C_4_H_8_O), which was used as a solvent, and benzoyl peroxide (BPO) (C_14_H_10_O_4_), which was used as an initiator. Ten milligrams of MMA monomer was added to 0.1 g of BPO initiator and THF was added as a solvent. The mixture was then left in a water bath for 24 h at a temperature of 80 °C under nitrogen gas protection. The polymer solution was purified by ethanol twice and left to dry.

### 3.2. PMMA-TiO_2_ Preparation

Chloroform was used to dissolve the PMMA polymer. TiO_2_ NPs (0.1 g) were dissolved in 10 mL of an ethanol and xylene mixture (50:50). Different concentrations of 0.05 and 0.1 mL of the prepared mixture were added to 10 mL of the PMMA polymer to obtain the doped PMMA-TiO_2_ nanocomposites with 0.005 and 0.01 *w*/*v*, respectively.

Scanning electron microscopy was performed by SEM 54032-GE02-0002/8038 (MIRA3/Austria) Austria. Measurement of optical properties of the prepared nanocomposites were carried out by UV-vis transmission spectroscopy using a UV-1601 PC (Shimadzu/Japan) Tokyo 101 Japan. The refractive index was measured using an Abbe refractometer at wavelengths 486.1, 587.6, and 656.3 nm, whereas the Abbe number was calculated using
(4)$$\nu_{d}=\frac{\left(n_{d}-1\right)}{\left(n_{f}-n_{c}\right)}$$
where *ν_d_* is the Abbe number and *n_c_*, *n_d_*, and *n_f_* are the RI values of the polymer films at wavelengths 656, 589, and 486 nm, respectively.

### 3.3. Optical Modeling

ZEMAX optical design software was used for modeling the human eye and the manufactured PMMA-TiO_2_ CLs. The evaluation of these lenses and image simulations were performed at a five-degree field of view (FOV) and at the photopic spectrum (white light) of 470, 510, 555, 610, and 650 nm wavelengths with weights of 0.091, 0.503, 1, 0.503, and 0.107, respectively. The Liou and Brennan eye model (LBEM) [24] was chosen (see Table 1) to evaluate the CLs’ effect. The anterior and posterior corneal surfaces were selected as aspherical surfaces. The pupil diameter was decentered nasally by 0.5 mm [25,26] and set at 4 mm.

The prepared aspherical CLs were constructed by inserting an extended polynomial surface as the front surface of the applied CL. The front surface radius of the CLs was set at 7.748 mm and its conic at 0.035 while the back surface was set at 7.8 mm and the prepared CL thickness was set at 0.1 mm.

## 4. Results and Discussion

Pure PMMA CL and 0.005 and 0.01 *w*/*v* PMMA-TiO_2_ CLs were prepared using a cast molding method and are shown in Figure 1. All of the synthesized nanocomposites were transparent, thin, and flexible, and were fabricated with a 0.1 mm thickness and 12 mm diameter

### 4.1. Morphological Properties

Scanning electron monographs of the prepared hybrid PMMA-TiO_2_ nanocomposites containing 0.05 and 0.1 wt. of TiO_2_ NPs are shown in Figure 2. From the SEM images, different sizes and shapes of TiO_2_ NPs can be seen to be imbedded in the PMMA polymer sample. The TiO_2_ NPs appear as bright points which are well distributed on the PMMA surface; this good distribution helped to improve the behavior of the prepared doped nanocomposites.

### 4.2. UV-Vis Transmission Spectrum

UV-vis transmission spectra of all the nanocomposites are shown in Figure 3 and indicate that the prepared polymer nanocomposites are highly transparent in the visible range. Transmittance of all films is higher than 95%, indicating the homogeneity of hybrid nanocomposites and the compatibility between the PMMA matrix and the inorganic metal. Figure 3 shows that pure PMMA transmits about 98.71% visible light, and due to the effect of doping with TiO_2_ NPs, this percentage was reduced to 98–96%. However, the prepared nanocomposites were transparent and maintained a value of transmittance above 95% in the visible region. The observed higher transmittance for the prepared PMMA-TiO_2_ nanocomposites is much better than that which has been obtained in some previous investigations [27,28].

### 4.3. Refractive Index and Abbe Number

High refractive index CLs will greatly improve ocular biocompatibility. Figure 4 illustrates the variation of refractive indices with different TiO_2_ contents. The refractive indices were measured for the prepared hybrid films at 486.1, 587.6, and 656.3 nm wavelengths. The variation of the refractive index with the wavelength components ascribes to the dispersion phenomenon. Each component is refracted by a specific refractive index through the sample. As a result of refractive index variation, the prepared hybrid nanocomposite dispersion values (*ν_d_*) were calculated from Equation (4) and are listed in Table 2. The index of refraction of the prepared CLs increased with increasing TiO_2_ NP concentration due to the polymer density increment (*n* = *c*/*v*), whereas the *ν_d_* values exhibit an opposite trend. Such RIs are superior to commercial CLs (1.43) as well as those which have been reported by others [29,30]. Doping with NPs of a higher refractive index than the pure PMMA refractive index increases the refractive index of the nanocomposites. For high optical quality, the particle size must be as small as possible to avoid the scattering effect.

### 4.4. Polychromatic MTF

The retinal image sharpness (spatial frequency value) and contrast (MTF value) are characterized by MTF criteria. For optimal retinal image quality, the lens should perform over 50% (0.5) contrast at 20 cycles/mm. The maximum spatial frequency was set at 30 cycles/mm as the optimum contrast area. Five-degree off-axis polychromatic MTF simulations were obtained for the simulated CLs compared to the free eye. A polychromatic MTF simulation is presented in Figure 5 which clearly shows that all of the hybrid CLs exhibited high image contrast at low frequencies (less than 20 cycles/mm), while the contrast value was degraded over 20 cycles/mm. The best vision correction was realized when 0.01 PMMA-TiO_2_ CL was used; for PMMA-TiO_2_ CL the area under the MTF curve was as large as possible. The other CLs made of pure PMMA and 0.005 PMMA-TiO_2_ exhibited almost similar behavior; vision was not corrected when they were applied but worsened. This is evidence that the image performance had been affected by refractive index variation. Although the doping process was done with the addition of small amounts of TiO_2_ NPs, it appeared to have a clear impact on the refractive index and thus on prepared hybrid CL efficiency.

Pure organic PMMA CL with RI = 1.491 and *ν_d_* = 53.32 achieved the worst MTF indication at 50%. On the contrary, the best contrast was obtained when the polymer was doped with 0.1 mL of TiO_2_ NPs, for which the MTF value was improved above 50% due to its optimum refraction properties and high transmissivity in visible light. There was a significant difference between pure PMMA and the doped PMMA. Where the difference in the contrast and sharpness was noteworthy was at the 50% and 20 cycles/mm intersection point. White light significantly degraded the visual quality of all cases. This degradation can be attributed to the presence of chromatic aberration in addition to the monochromatic aberrations in the eye, i.e., spherical, coma, astigmatism, and distortion aberrations [31]. Chromatic aberration results from the separation of white light into its wavelength components and the focusing of these components on different focal points [32]. These aberrations were reduced when the high TiO_2_ content CL was inserted on the eye.

### 4.5. Spot Diagram

Spot diagrams of an LBEM eye of *RMS* = 3.295 and treated eyes with different CL concentrations are presented in Figure 6. When pure PMMA and 0.005 PMMA-TiO_2_ CLs were inserted, the highest spot sizes were obtained and *RMS* was 7.518 and 4.206 µm, respectively. These CLs enlarged the spot sizes and increased their *RMS* by more than the LBEM eye because chromatic aberrations further reduced the image quality in addition to monochromatic aberrations (spherical, coma, and astigmatism aberrations). The image deformation was treated by 0.01 PMMA-TiO_2_ CL, spot size was minimized, *RMS* = 2.738 µm, and a high image performance was obtained.

### 4.6. Image Simulation

A formed image of each model is shown in Figure 7. 0.01 PMMA-TiO_2_ CL revealed the best image clarity with polychromatic light sources. The images formed by the prepared lenses without NP addition (PMMA-only) and using 0.005 PMMA-TiO_2_ CLs were blurred and worse than the retinal image formed by the eye without CL. The generated spherical, coma, and astigmatism aberrations prevented the formed image from being sharp where the rays from the object point did not focus into a single focal point. There was no noticeable improvement in image contrast and this proved what has been discussed in the MTF analysis. Only the highest refractive index CL carried out the best image correction.

## 5. Conclusions

Nanomaterial-doped CLs are considered promising devices for vision correction. The obtained results of our PMMA-TiO_2_ CL programming test are encouraging. The aim of this work was to investigate TiO_2_-doped CL performance on the aberrated eye when the human eye operates under polychromatic light sources. The present study exhibited a closer MTF response to the diffraction-limited MTF of a healthy eye by the use of the mentioned CL. Regardless of cut-off frequency value, the highest image contrast was realized at the 0.1 wt. TiO_2_ concentration. The generated monochromatic and polychromatic aberrations were overcome. Spherical aberration (SA) cannot be altered by CL bending since its shape must be fixed to comply with the corneal shape. In spite of this, the applied aspheric front surface CL minimized the SA level and did not eliminate it by focusing the edge and central rays in a small spot size on the retinal surface. This CL reduced both the axial color defocus and the lateral color tilt produced by the polychromatic light source. The TiO_2_ NPs were well distributed in the PMMA polymer, as seen in the SEM images. In addition, the prepared CL was characterized by a high RI value of about 1.615, low dispersion (*ν_d_* = 31), and high transparency in the visible region (T > 95%).

## Figures and Tables

**Figure 1 nanomaterials-10-02028-f001:**
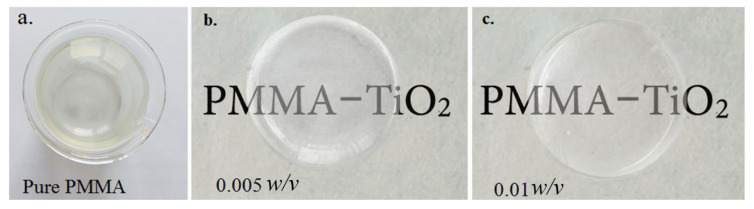
Photographs of synthesized samples. Legend: PMMA, poly(methyl methacrylate).

**Figure 2 nanomaterials-10-02028-f002:**
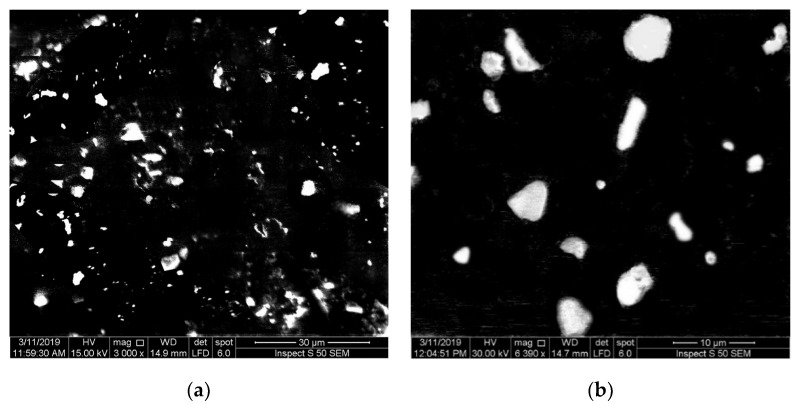
SEM monograph of 0.01 *w*/*v* PMMA-TiO_2_ samples (**a**) 0.05 and (**b**) 0.1 wt.

**Figure 3 nanomaterials-10-02028-f003:**
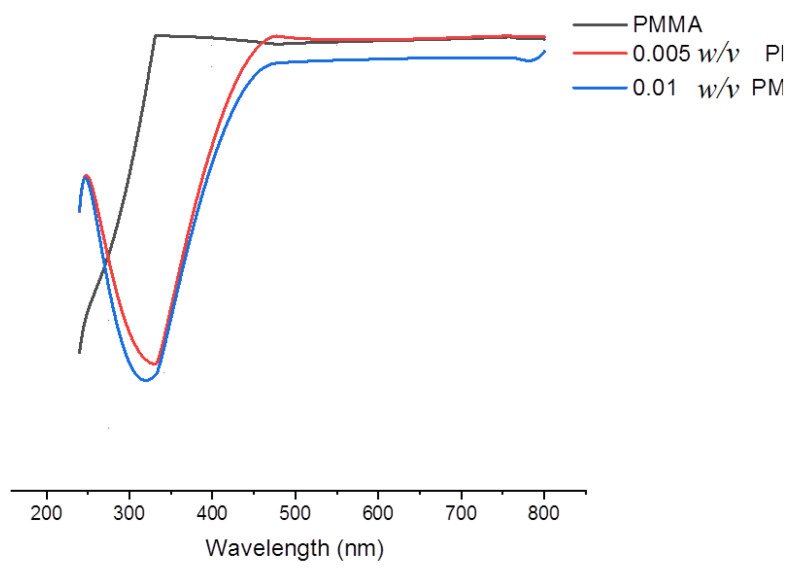
UV-vis transmission spectrum of pure PMMA and 0.005 and 0.01 PMMA-TiO_2_.

**Figure 4 nanomaterials-10-02028-f004:**
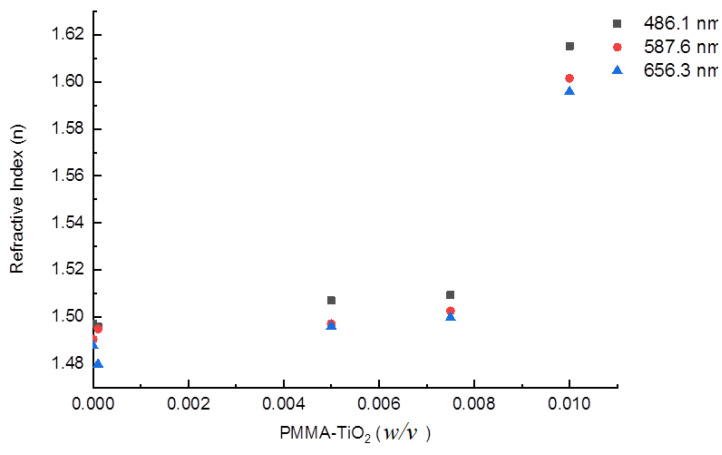
Variation of refractive index with TiO_2_ content at specific wavelengths.

**Figure 5 nanomaterials-10-02028-f005:**
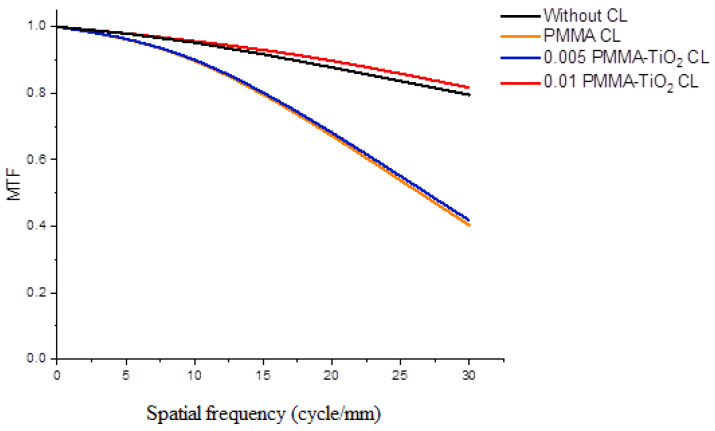
Polychromatic modulation transfer function (MTF) simulation curves of prepared contact lenses (CLs) in compare with a free eye.

**Figure 6 nanomaterials-10-02028-f006:**
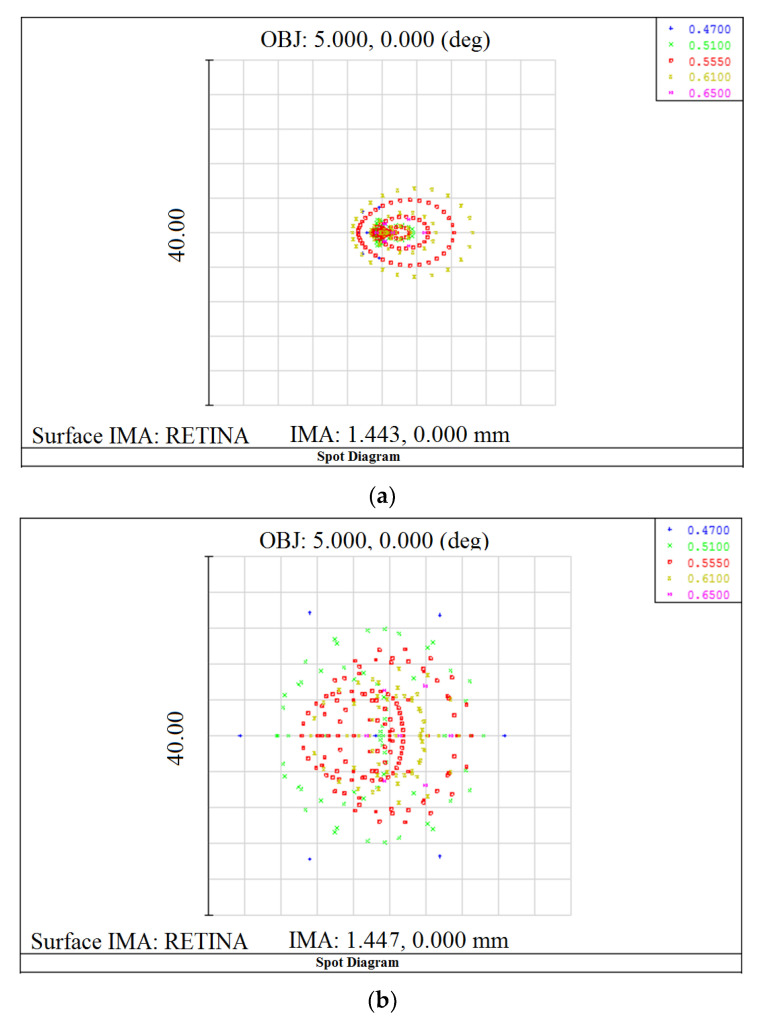
Comparison of the formed retinal image (**a**) without CL, (**b**) with PMMA CL, (**c**) with 0.005 PMMA-TiO_2_ CL, and (**d**) with 0.01 PMMA-TiO_2_ CL.

**Figure 7 nanomaterials-10-02028-f007:**
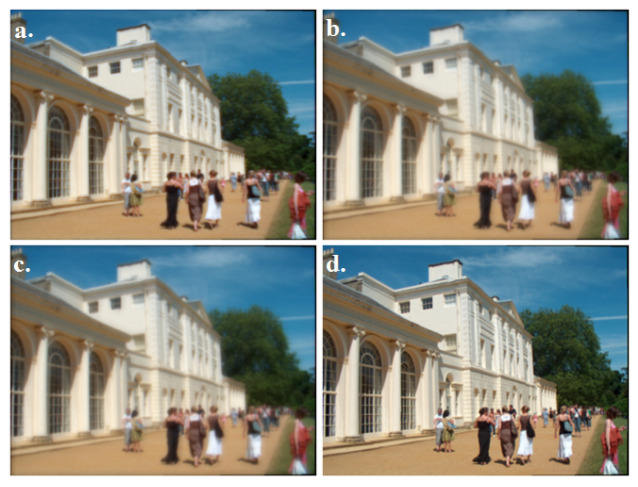
Human eye retinal image simulation (**a**) without CL, (**b**) with PMMA CL, (**c**) with 0.005 PMMA-TiO_2_ CL, and (**d**) with 0.01 PMMA-TiO_2_ CL.

**Table 1 nanomaterials-10-02028-t001:** Input parameters of the Liou and Brennan eye model (LBEM) in ZEMAX software [24]. Legend: RI, refractive index; PD, pupil diameter; *v_d_*, Abbe number.

Surface	Radius (mm)	Thickness (mm)	RI	*ν_d_*	Conic	PD (mm)
Cornea (anterior cornea)	7.77	0.55	1.376	50.23	−0.18	10
Aqueous (posterior cornea)	6.40	3.16	1.336	50.23	−0.6	10
Pupil	Infinity	0	1.34	50.23	0	4
Lens—front surface	12.40	1.59	-	-	0	10
Lens—back surface	Infinity	2.43	-	-	0	10
Vitreous humor	−8.1	16.24	1.336	50.23	0.96	10
Retina	−12	-	-	-	0	10

**Table 2 nanomaterials-10-02028-t002:** Refractive index and Abbe number of the prepared samples.

PMMA-TiO_2_ (*w/v*)	486.1 nm	587.6 nm	656.3 nm	*ν_d_*
0	1.497	1.491	1.488	53.32
0.005	1.507	1.497	1.496	45.61
0.01	1.615	1.602	1.596	31.00

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
