# Peer review of "Manufacture of Contact Lens of Nanoparticle-Doped Polymer Complemented with ZEMAX"

_nanomaterials, 2020, doi:10.3390/nano10102028_

Round 1
Reviewer 1 Report
This work discusses the preparation of of TiO2 NPs doped PMMA films for their applied as contact lenses. The authors have evaluated the hence derived films for their ocular properties by refractive index (RI) based calculations and subsequent image simulation. The manuscript is well written and presents a possibility of using NP doped polymers to improve the RI for better contrast and visibility. The manuscript is well written and presents a good perspective. However some issues should be addressed in detail:
1. The authors must mention the source of the raw materials for TiO2 particle preparation. What is the average size/size distribution of the TiO2 nanoparticles used in this study. In figure 2, from the SEM figures it seems like the particles clump to form dense >1 micrometer scale structures.
2. The cast moulding methods needs more information about the kind of cast and perhaps the dimensions. Were the lens thickness and curvature defects/errors identified after cast moulding?
3. Have the authors performed Thermogravimetric/ Calorimetric experiments to assess the threshold melting temperature or glass transition (Tg) temperature of the prepared composites?
4. The authors could discuss more about the possible mechanisms and reasons to observe higher refraction obtained due to increased doping.
5. The number of experiments performed (on independent lenses) in determining refractive index and the abbe numbers are not mentioned since perhaps a single average value is shown in the tables.
6. The axes, legends and the chosen wavelengths is not clear in figure 6. The authors need to elaborate their discussion of this figure and make a thorough comparison of the trends in the rms value distributions to accurately assess and compare the image sharpness in each case
Author Response
Dear reviewer,
Thank you for your comments and suggestions, all have been done point-by-point so, please see the revised manuscript.
Please see the attachment

Reviewer 2 Report
Manuscript: Manufacture of contact lens of nanoparticle doped polymer complemented with ZEMAX
Manuscript ID: nanomaterials_635804
Comments: This manuscript presented the preparation of PMMA-TiO2 nanocomposite as a new, promising contact lens material. SEM, UV-Vis transmission spectrum, and MTF was applied to characterize the physical and optical properties of the synthesized nanocomposites, respectively. The authors also proved that 0.01 PMMA-TiO2 nanocomposite has the largest refractive index, lowest RMS in spot diagram, and the best image clarity in eye retinal image simulation. Thus, I believe this study could contribute the discovery of novel contact lens. I would recommend the acceptance of this manuscript after minor revision.
A few concerns are listed as follows:
The language needs to be carefully polished. I observed there were many grammar errors in the manuscript. For example, on line 30, “a polymer that fulfill …” should be “a polymer that fulfills …”; line 32, “CLs classified into …” should be “CLs were classified into …”; line 43, “adversely effect on …” should be “adversely affect …” or “have an adverse effect on …”. I highly recommend that the authors should go through the entire manuscript carefully to check these errors.Figure 5 is not that clear since some line colors were very close. I suggest using some other colors instead.
Figure 4 showed that the higher content of TiO2 in PMMA-TiO2 always leads to higher refractive index, and lower dispersion. If the concentration of TiO2 keeps increasing beyond 0.01 we/vol, could you still observe ever-increasing RI values?
In Figure 6 and Figure 7, it was shown that only with 0.01 PMMA-TiO2 CL the better retinal images were achieved, while PMMA CL and 0.005 PMMA-TiO2 CL performed even worse than the one without CL. Does that mean the PMMA-TiO2 can’t improve the vision correction when the TiO2 concentration is low? Can you get better performance by continuous increasing the TiO2 concentration?
Author Response

(The authors gave the same response as above.)
